# High Variability in Human Sperm Membrane Potential over Time Can Limit Its Reliability as a Predictor in ART Outcomes

**DOI:** 10.3390/biology14070851

**Published:** 2025-07-12

**Authors:** Tomás J. Steeman, Carolina Baro Graf, Analia G. Novero, Mariano G. Buffone, Dario Krapf

**Affiliations:** 1Instituto de Biología Molecular y Celular de Rosario (IBR CONICET–UNR), Rosario 2000, Argentina; steeman@ibr-conicet.gov.ar (T.J.S.); barograf@ibr-conicet.gov.ar (C.B.G.); novero@ibr-conicet.gov.ar (A.G.N.); 2Instituto de Biología y Medicina Experimental (IBYME–CONICET), Buenos Aires 1428, Argentina; buffone@dna.uba.ar

**Keywords:** membrane potential, sperm capacitation, human fertility

## Abstract

Hyperpolarization of the sperm membrane potential (*E*m) during capacitation is a key marker of fertilizing ability. In human sperm, samples that succeed in conventional in vitro fertilization displayed hyperpolarized *E*m values. Thus, *E*m has been proposed as a predictive biomarker for in vitro fertilization (IVF) success. We evaluated whether membrane potential stays consistent over a period of weeks and could so be employed to predict if upcoming IVF procedures will work. We measured membrane potential in capacitated and non-capacitated sperm samples from 18 healthy donors at three points each over a period of 28 days, and we found that samples failing to hyperpolarize beyond a previously determined threshold linked to IVF success show high variability within individuals. This variability was also higher after capacitation than among basal conditions. Due to this variability, measuring membrane potential in advance may not reliably predict fertility outcomes. However, it remains a promising functional marker when assessed on the day of IVF. Our results underscore the importance of timing when using functional tests to assess sperm quality. Understanding why this variability happens could help improve fertility testing and treatment decisions, potentially benefiting couples undergoing assisted reproduction.

## 1. Introduction

Infertility is a widespread global health issue affecting approximately one in seven couples, or roughly 80 million people worldwide [1,2]. The causes of infertility are diverse, with male factors now recognized as equally important as female factors, accounting for at least 50% of cases [3]. Couples struggling with infertility often turn to assisted reproductive technologies (ARTs) to conceive, which include intra-uterine insemination (IUI), in vitro fertilization (IVF), and intra-cytoplasmic sperm injection (ICSI). While these treatments have made significant advancements, they are costly, invasive, and carry inherent risks. The decision to pursue ART is often based on general sperm parameters, measured in their basal, non-capacitated state. However, these basal measurements do not necessarily predict treatment success. Therefore, a deeper understanding of sperm behavior, particularly the physiological and molecular processes that contribute to fertilizing ability, is essential for the development of more accurate diagnostic tools and improved ART decision-making [4,5].

During fertilization, sperm fuses with the oocyte to deliver its genetic contents. However, mammalian sperm must first undergo a series of physiological and morphological changes, known as ‘capacitation’, to acquire fertilizing ability. Capacitation involves a complex sequence of events that typically occurs within the female reproductive tract but can be mimicked in vitro in defined media. This process promotes changes in motility (referred to as hyperactivation) and primes the sperm to undergo the acrosome reaction in response to appropriate stimuli [6,7].

At the molecular level, sperm capacitation is associated with an increase in membrane fluidity, changes in intracellular ion concentrations [8], hyperpolarization of the sperm plasma membrane potential (*E*m) [9,10], increased protein kinase A activity [11], increase in intracellular Ca^2+^ concentrations [12], and pH alkalinization [13]. These events collectively contribute to the acquisition of fertilizing ability. Among them, the change in *E*m, mediated by K^+^ channel SLO1 in human sperm [14], has been shown to be central to the process of capacitation. Male mice devoid of Slo3, the K^+^ channel responsible for hyperpolarization in this species, are infertile [15]. The essence of membrane hyperpolarization lies in the difference between the ionic concentration of the intracellular and extracellular environments. Prior to capacitation, the balance between ion fluxes, gradients, and permeabilities results in an electric potential known as resting *E*m [8]. As mammalian sperm traverse the female reproductive tract, they encounter environments with markedly different extracellular compositions and must adapt by regulating their *E*m. This regulation is important, as ion channels and transporters, like the sperm-specific Ca^2+^ channel CatSper and the voltage-gated proton channel Hv1, are affected by the membrane potential [16,17].

Capacitation-associated *E*m changes have been thoroughly studied in mouse sperm, where it is both necessary and sufficient to promote the acrosome reaction [18]. In human sperm, several studies report a resting *E*m of −40 ± 16 mV, whereas after 5 h of capacitation, sperm showed an *E*m −58 ± 2 mV [19,20]. Previous independent studies from our lab and others established a correlation between hyperpolarization and IVF success, with an *E*m of at least −48.6 mV or −46 mV identified as necessary for successful in vitro fertilization [21,22]. It is important to note that these thresholds were derived from studies involving clinic patients whose sperm samples were classified as normozoospermia according to WHO guidelines, as they were undergoing conventional IVF procedures. These findings suggested that *E*m measurements could be used as predictors of assisted reproductive treatment through a simple fluorometric assay [23], aiding the decision between different ART methods (such as IVF or ICSI). Worth noticing, these evaluations have been performed on the same sperm samples used for ART. Thus, by the time of the result, it might be too late to choose the best ART method.

Here, we aimed to analyze whether *E*m measurements performed prior to the treatment day could assist in the decision-making process. We measured membrane potential in 108 conditions of 54 sperm samples from 18 normospermic donors over the span of 28 days and evaluated the variation over time. Our results reveal substantial temporal variability in sperm membrane potential upon capacitation among the same donors. Therefore, caution should be taken when considering using *E*m-based prediction of ART success.

## 2. Materials and Methods

### 2.1. Ethics Approval

Volunteer donors were provided with written information about the study prior to giving informed consent. The study protocol was approved by the Bioethics Committee of the Facultad de Ciencias Bioquímicas y Farmacéuticas, Universidad Nacional de Rosario, protocol #564/2018. The studies are in compliance with the Declaration of Helsinki principles.

### 2.2. Sperm Culture Media

HEPES-buffered human tubal fluid (HTF, NaCl 90.7 mM, KCl 4.7 mM, CaCl_2_ 1.6 mM KH_2_PO_4_ 0.3 mM, MgSO_4_ 1.2 mM, HEPES 23.8 mM, D-Glucose 2.8 mM, sodium pyruvate 3.4 mM, sodium lactate 21.4 mM, pH 7.4) was prepared in-house using analytical-grade reagents from Cicarelli (San Lorenzo, Argentina), Sigma-Aldrich (St. Louis, MO, USA) and Merck (Darmstadt, Germany), and used as ‘non-capacitating media’ (NC). HTF supplemented with NaHCO_3_ 15 mM (Sigma-Aldrich) and BSA 0.5% *w*/*v* (A7906, heat-shock fraction, ≥ 98%, Sigma-Aldrich) was used as ‘capacitating media’ (CAP).

### 2.3. Human Sperm Preparation

Semen samples were obtained by masturbation from healthy donors after 2–5 days of abstinence. All donors were under 40 years of age, reported only occasional alcohol consumption, had no drug use, had no history of varicocele, and were not taking any hormone-related medication. Samples that fulfilled semen parameters according to WHO recommendations (including assessments of ejaculate volume, sperm concentration, total sperm number, motility, morphology, and agglutination—see Appendix A for recorded parameters) [24] were allowed to liquify for 1 h at 37 °C in a water bath. Then, spermatozoa were separated by two-layer density gradient centrifugation (PureSperm 40/80, Nidacon, Sweden) and washed with HTF.

### 2.4. Em Determination by Fluorometric Population Assay

Conditions were prepared in 400 µL of either NC or CAP medium at a sperm concentration of 4 × 10^6^ cells/mL. After 5 h of incubation, cells were loaded with DiSC_3_(5) 1 µM (Molecular Probes, Thermo Fisher Scientific, Waltham, OR, USA) in a final volume of 1.7 mL HTF and transferred to a gently stirred quartz cuvette at 37 °C. Fluorescence was monitored with a Varian Cary Eclipse fluorescence spectrophotometer (Palo Alto, CA, USA) at 620/670 nm excitation/emission wavelengths. Recordings were initiated when steady-state fluorescence was reached (approximately after 10 min). Calibration was performed by adding 1 µM valinomycin (Cayman, 10009152, Ann Arbor, MI, USA) and sequential additions of KCl as previously described [23]. Finally, sperm membrane potentials were obtained by linearly interpolating the theoretical *E*m values against arbitrary fluorescence units of each trace. The theoretical *E*m values were obtained using the Nernst equation, considering 120 mM as the internal K^+^ concentration in sperm. This internal calibration curve for each determination compensates for variables that influence the absolute fluorescence values.

### 2.5. Statistical Analysis

Data are expressed as mean of *E*m ± standard error of the mean (SEM), for each measurement per donor, as indicated in each figure. Statistical significance was set at *p* < 0.05. Statistical analyses were performed using the GraphPad Prism 9 software (La Jolla, CA, USA) and Python (version 3.11.4), with thepingouin (version 0.5.5) and scipy (version 1.14.1) libraries. Plots were generated using the matplotlib (version 3.9.3) and seaborn (version 0.13.2) libraries.

## 3. Results

To investigate whether sperm membrane potential (*E*m) measurements taken sequentially over one month could serve as reliable predictors of IVF success, we measured *E*m values in sperm samples from 18 different donors. Measurements were performed on basal non-capacitated state sperm (NC, i.e., incubated in the absence of bicarbonate and albumin, Appendix A) and after 5 h in capacitating conditions (CAP, i.e., in the presence of bicarbonate and albumin). The 5 h time point was selected based on previous studies showing that cells start to hyperpolarize at 3 h of capacitation, with *E*m further stabilizing and polarizing by 5 h [22]. We performed these measurements for each donor at day 0, day 14, and day 28 (Figure 1). The threshold for membrane hyperpolarization was set at −48.6 mV for *E*m_CAP_, as identified by Baro Graf, et al. (2020) [22], since a depolarized cell membrane was associated with IVF failure in idiopathic subfertile patients. Donors with at least one measurement failing to hyperpolarize beyond this threshold are indicated with asterisks.

Results indicate that a subset of donors (donors 3, 5, 7, 8, 9, 10, 12, 13, 14, and 15) exhibited at least one depolarized sample upon capacitation, with *E*m values above −48.6 mV. In contrast, donors such as 1, 2, 4, and 17 consistently displayed membrane potentials well below the threshold during the whole period. Notably, some donors that failed to hyperpolarize past the threshold in at least one time point exhibited highly hyperpolarized membranes on different days, emphasizing the variability of sperm membrane potential. Overall, low hyperpolarization events were more frequently observed in donors with higher variability, as indicated by the larger dispersion associated with these individuals.

While some samples exhibited a hyperpolarized membrane potential upon capacitation, this was not consistent, accounting for the variability previously seen in human samples. In addition, when comparing membrane potential across donors, we observed significant inter-donor variability. Consistency within donors was observed in samples exhibiting hyperpolarized values (1, 2, 4, 6, 11, 16, 17, and 18). However, in other donors, *E*m values varied substantially between samples, and no consistent depolarization pattern was observed across timepoints. To statistically evaluate the variation in *E*m values, we performed an ANOVA analysis of repeated measures. The analysis revealed a statistically significant effect of time over *E*m (*p* = 0.0076, η^2^_G_ = 0.107), indicating that *E*m does not remain stable over the 28-day period. The largest difference was found between day 0 and day 28 (days 0–14: p_corr_ = 0.054; days 0–28: p_corr_ = 0.028; days 14–28: p_corr_ = 1; Bonferroni post hoc comparison).

In light of these results, we aimed to assess the relative variability using the coefficient of variation (CV) as a statistical method. CV, defined as the ratio of the standard deviation to the mean of a dataset, allows for comparison of the degree of variation from one donor dataset to another, even when the means are drastically different. The CV was calculated for each donor in both non-capacitated (NC) and capacitated (CAP) subsets of sperm samples (Figure 2A). As a threshold for high variability, a cutoff of 30% CV was established (dotted line) as an exploratory indicator to identify donors exhibiting substantial intra-individual biological fluctuation in *E*m. This magnitude of variability is considered significant given the known high biological variation of other semen parameters [25] and the typically lower analytical variability of quantitative assays [23,26]. Donors exceeding this threshold in at least one condition are indicated with asterisks, indicating high variability in *E*m results along measurements. These results suggest that the outcome of an *E*m measurement on a given day may not reliably predict the potential success of an IVF attempt performed two weeks later with a new sperm sample.

Overall, capacitated sperm exhibited higher CVs compared to non-capacitated sperm, with several donors (e.g., 5, 8, 12, 13, and 14) showing substantial variability after 5 h of capacitation. In contrast, non-capacitated samples generally displayed more stable measurements, with a few exceptions of high variability (e.g., donors 10 and 12). Notably, some donors with high CVs in the capacitated condition also failed to hyperpolarize beyond the −48.6 mV threshold (e.g., donors 8, 12, and 13), suggesting a potential link between hyperpolarization and variability. However, Pearson regression tests only showed statistical correlation between high CV and higher mean membrane potential for the non-capacitated samples, and not for capacitated samples (Figure 2B,C, *p* = 0.018 for non-capacitated sperm, *p* = 0.51 for capacitated sperm).

To further explore the potential factors influencing *E*m variability, we analyzed correlations between *E*m values after capacitation and semen volume, sperm number, and sperm concentration measured at the same time point. We found a statistically significant, albeit weak, correlation between hyperpolarization and sperm concentration (Appendix A). No significant correlations were observed for total sperm number or semen volume (Appendix A).

Together, these findings highlight significant inter-donor variability in sperm *E*m among samples taken along a 2 week interval time. Capacitated sperm generally exhibited a greater variability than that of non-capacitated sperm. These results suggest that caution should be taken when using sperm membrane potential measurements in advance of an IVF treatment as a predictor of sperm fertilizing capacity. Further validation is needed to determine their reliability to predict success rate in IVF procedures not performed on the same day.

## 4. Discussion

A total of 108 conditions were analyzed in 54 normospermic sperm samples from 18 donors to determine whether membrane potential (*E*m) remains stable over three different time points taken over 28 days. We found that *E*m varied significantly within individuals over time. In fact, around 50% of the donors exhibited changes in *E*m large enough to cross a previously proposed functional cutoff of 30% CV, which would result in inconsistent sample classification [22].

This finding is particularly relevant considering previous work demonstrating that capacitation-associated hyperpolarization is linked to the acquisition of fertilizing capacity in human sperm [21,22]. Those studies reached similar conclusions independently using different experimental approaches. Baro Graf et al. [22] used population fluorimetry to show that capacitated samples showing *E*m below −48.6 mV had increased success rates. On the other hand, Puga-Molina et al. [21] used flow cytometry to also show that samples exhibiting *E*m values below −46 mV had increased success rates. In addition, the degree of *E*m hyperpolarization after capacitation correlated strongly with acrosomal responsiveness [21,22]. Altogether, these results strongly suggested that *E*m could be applied as a predictive functional biomarker. However, our results reveal that *E*m cannot be considered a fixed trait within individuals, at least when measured in different ejaculates collected weeks apart under standardized conditions.

This temporal variability highlights an important limitation for the implementation of *E*m as a standalone predictive marker in assisted reproduction. While *E*m remains a strong correlate of fertilizing competence when measured immediately before IVF [21,22], our data suggest that measurements taken weeks in advance may not reliably represent the fertilizing potential of future ejaculates from the same donor. In ART, ICSI is widely used, frequently even when normospermic profiles are present, effectively bypassing many aspects of sperm function. This often stems from a clinical perception of higher success rates or to circumvent potential IVF failures. However, for normospermic cases, conventional IVF is still considered standard. Tools that help predict IVF success and refine the choice between IVF and ICSI are thus invaluable for optimizing outcomes and avoiding potentially unnecessary interventions and costs. Therefore, tests that assess sperm fertilizing ability, such as *E*m measurements, retain clinical relevance in aiding the choice between IVF and ICSI in these patients. In this context, it becomes crucial to define the optimal time window for functional sperm assessments intended to guide clinical decisions.

The reasons behind this variability in *E*m remain to be elucidated. It is possible that inter-ejaculate differences in semen composition for the same individual, hormonal levels [27,28], or testicular environment (factors known to fluctuate over time even within healthy individuals) could influence the capacitation response and, consequently, *E*m. Moreover, although capacitation protocols were identical across all time points, the biological response to those conditions may vary depending on the physiological state of the sperm at the time of collection. These subtle biological fluctuations, while insufficient to alter conventional semen parameters, may be enough to impact functional outputs like membrane potential. While we found a weak correlation between hyperpolarization and initial sperm concentration, this finding alone does not fully explain the large temporal fluctuations in *E*m observed, nor does it stablish a causal relationship.

Several physiological and environmental factors could underlie the temporal fluctuations in sperm membrane potential observed in our study. Hormonal cycles, particularly fluctuations in testosterone and FSH levels, may influence spermatogenesis and epididymal maturation, subtly affecting the capacitation response [29]. Oxidative stress is another candidate mechanism: even in normozoospermic men, reactive oxygen species (ROS) levels can vary between ejaculates due to factors such as diet, physical activity, or transient inflammation. ROS can affect membrane lipid composition and ion channel function, both of which are critical for *E*m regulation [30,31]. Additionally, circadian and seasonal rhythms in testicular function, as recently described in healthy men, might contribute to these dynamics [32]. Although our donors were screened for known confounders (e.g., varicocele, illness, and medications), lifestyle factors such as stress or sleep patterns could also affect sperm physiology. Ultimately, our findings highlight the critical need for further mechanistic studies to determine the underlying causes of this inter-individual variability in *E*m, opening significant new avenues for research in sperm physiology and male fertility. Future studies including hormonal profiling, ROS quantification, and broader phenotyping may help elucidate the mechanisms governing *E*m variability.

While our study focused on normozoospermic donors to evaluate the inherent temporal variability of *E*m under standard conditions, further investigation is needed to assess how this variability behaves in samples with altered semen parameters. Given the added complexity of oligo-, astheno-, or teratozoospermic profiles, understanding *E*m dynamics in these populations could provide additional insights into their fertilization potential and guide ART strategies.

Altogether, our findings provide novel insight into the temporal dynamics of human sperm membrane potential and raise important questions regarding the physiological factors influencing functional sperm parameters over time. Further studies will be necessary to determine the underlying causes of *E*m variability and to assess whether specific patterns of *E*m fluctuation are associated with fertility outcomes.

## 5. Conclusions

Our study reveals that human sperm membrane potential (*E*m) is not stable over time, even when measured under standardized capacitation conditions. Across three time points spanning 28 days, *E*m varied significantly within individuals, with approximately half of the donors exhibiting intra-individual variability exceeding a 30% coefficient of variation threshold. These fluctuations were sufficient to alter sample classification relative to a previously proposed functional *E*m cutoff, raising concerns about the reliability of *E*m as a long-term predictive biomarker.

These findings underscore a critical limitation in using *E*m measurements taken days or weeks before assisted reproduction procedures to assess sperm fertilizing potential. While *E*m remains a robust correlate of sperm function when measured immediately prior to IVF, our data suggest that its predictive value diminishes over time due to underlying biological variability. Further research is warranted to elucidate the physiological drivers of *E*m fluctuation and to determine whether specific *E*m dynamics are linked to fertility outcomes.

## Figures and Tables

**Figure 1 biology-14-00851-f001:**
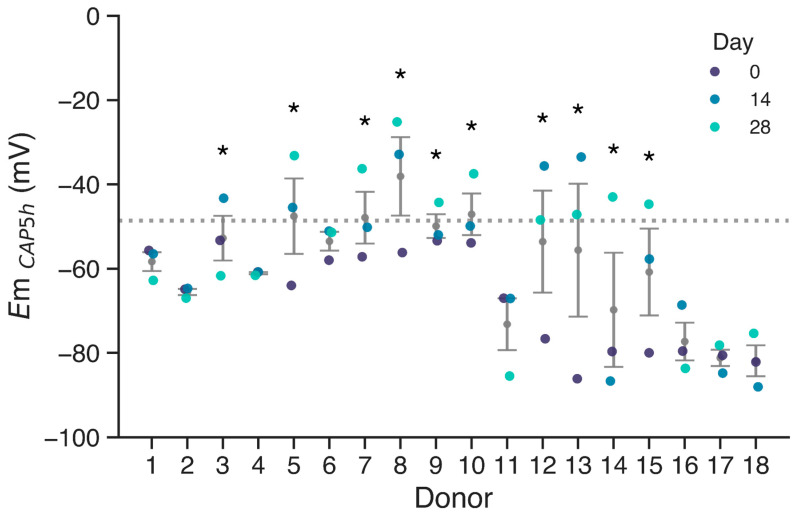
Membrane potential of sperm cells after 5 h of capacitation, grouped by donor. The −48.6 mV threshold is shown with the dotted line. Donors with at least one measurement over −48.6 mV are indicated with asterisks. Mean ± SEM in gray.

**Figure 2 biology-14-00851-f002:**
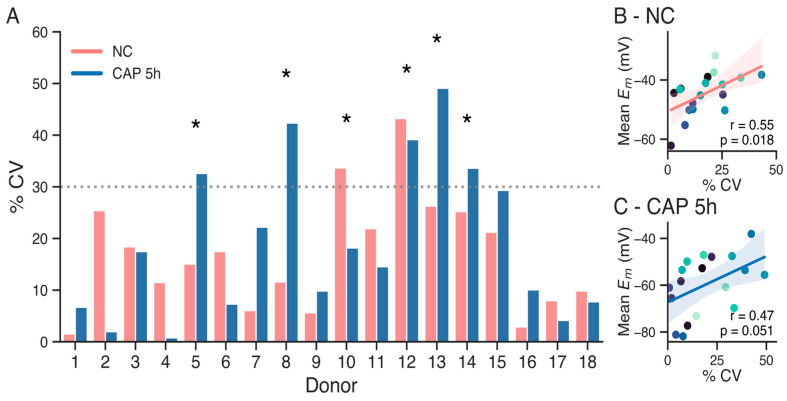
(**A**). Coefficient of variation between repeated measurements of non-capacitated sperm (pink), and after 5 h of capacitation (blue). The 30% CV threshold is shown with the dotted line. Donors with higher than 30% CV in at least one condition are indicated with asterisks. (**B**). Correlation between coefficient of variation and mean membrane potential of non-capacitated sperm. Pearson test, r = 0.55, *p* = 0.018. (**C**). Correlation between coefficient of variation and mean membrane potential of capacitated sperm. Pearson test, r = 0.47, *p* = 0.051.

## Data Availability

Data can be found in the Appendix A.

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
