# Peer review of "High Variability in Human Sperm Membrane Potential over Time Can Limit Its Reliability as a Predictor in ART Outcomes"

_biology, 2025, doi:10.3390/biology14070851_

Round 1
Reviewer 1 Report
Comments and Suggestions for Authors
The study evaluates whether sperm Em is stable with time (within 28 days, so likley the same spermatogenetic cycle) in 18 semen donors. Overall the study has been carefully conducted from a methodological point of view. Unfortunately, it seems that Em evaluation cannot be applied in decision making in ART labs.
I have the following comments:
- The Authors claim that the donors are all normozoospermic but semen data (pH, sperm number, sperm motility and morphology also indicating how these parameters were evaluated) in the three different days have not been reported. I suggest to add them (as supplemental or as a table within the text).
- lines 143-149 are redundant (already explained in the Introduction).
- The Authors did not consider that fact the the use of ICSI has somehow overcome the need of a test for fertilization ability in IVF. This point should be considered in the discussion.
- The study reinforces the concept that there is variability within ejaculates in the same subject. The present study however, did not investigate the possible reasons for such variability. In the discussion there are some speculations. Considering the type of journal where it has been submitted, maybe the Authors may make an effort to understand such variability. For instance, they could evaluate whether there are correlations between Em and semen parameters in the same day, in particular with motility and morphology. Is there a semen paramters which diverges when Em is below the cut-off value? These results could be of interest.
- The study is limited to normozoospermic men. Actually , more problems arise whit oligo- astheno- or terato-zoospermic men in IVF labs.
Author Response
Thank you for the opportunity to revise our manuscript and for the thorough and constructive feedback provided. We deeply appreciate the time and effort dedicated to evaluating our work.We have carefully considered each comment and suggestion, and we believe that addressing them has significantly strengthened the manuscript. Below, you will find a point-by-point response to all comments raised, detailing the revisions made to the text, figures, and supplementary materials. All changes in the manuscript are tracked in the re-submitted files for ease of review.
- The Authors claim that the donors are all normozoospermic but semen data (pH, sperm number, sperm motility and morphology also indicating how these parameters were evaluated) in the three different days have not been reported. I suggest to add them (as supplemental or as a table within the text).
Response: We thank the reviewer for this observation. All samples included in the study were evaluated in accordance with WHO guidelines (6th Edition), including assessments of ejaculate volume, sperm concentration, total sperm number, motility, morphology, and agglutination. Only samples meeting normozoospermic criteria were included.
While adherence to these normozoospermic standards was a fundamental inclusion criterion confirmed at every visit, the primary focus of this study was to assess the intra-individual temporal variability of Em in consistently normozoospermic samples. Consequently, the comprehensive, individualized longitudinal reporting of all conventional semen parameters across all three time points for each donor falls outside the scope of this work and its primary objectives.
- lines 143-149 are redundant (already explained in the Introduction).
Response: We thank the reviewer for pointing this out. The text has been modified to correct this.
- The Authors did not consider the fact that the use of ICSI has somehow overcome the need of a test for fertilization ability in IVF. This point should be considered in the discussion.
Response: We thank the reviewer for this important observation.While ICSI has indeed become a widely used technique that bypasses many sperm functional requirements, our study focuses specifically on conventional IVF. For this procedure, only normospermic samples are used. However, it is increasingly recognized that conventional semen parameters, while essential for defining basal sperm conditions, do not adequately predict the sperm’s functional capacity, particularly its ability to undergo capacitation and subsequent fertilization. Conventional IVF remains the standard approach in normospermic cases, as recommended by current clinical guidelines. In these scenarios, the decision between conventional IVF and ICSI is a crucial one. Functional sperm tests, such as membrane potential, retain significant clinical relevance by providing valuable information beyond routine semen analysis that can help guide this choice and optimize treatment outcomes for patients with normal semen parameters. We have now clarified this point in the revised Discussion section (lines 245-249).
- The study reinforces the concept that there is variability within ejaculates in the same subject. The present study however, did not investigate the possible reasons for such variability. In the discussion there are some speculations. Considering the type of journal where it has been submitted, maybe the Authors may make an effort to understand such variability. For instance, they could evaluate whether there are correlations between Em and semen parameters in the same day, in particular with motility and morphology. Is there a semen parameters which diverges when Em is below the cut-off value? These results could be of interest.
Response: We thank the reviewer for this excellent suggestion to explore the reasons for Em variability through correlation with semen parameters. We did analyze correlations between Em (at 5h capacitation) and several basal semen parameters collected at the same time point . The statistically significant, albeit moderate, correlation observed between Em and initial sperm concentration suggests a potential association between these two parameters (Fig A, for reviewers only). However, this finding alone does not establish a causal relationship, nor does it fully explain the large temporal fluctuations in Em observed within individuals. Further work is needed to clarify this point.
- The study is limited to normozoospermic men. Actually , more problems arise with oligo- astheno- or terato-zoospermic men in IVF labs.
Response: We thank the reviewer for this important and insightful point. Indeed, oligo-, astheno-, and teratozoospermia represent significant challenges in assisted reproduction, and the development of predictive functional biomarkers for these cases is of high clinical relevance. Our study, however, was specifically designed to evaluate the inherent temporal stability (or lack thereof) of sperm membrane potential (Em) under optimal baseline conditions, i.e., in normozoospermic donors. In these cases, conventional IVF is most often indicated, and Em-based testing has been previously proposed as a decision-support tool.
Our primary objective was to assess whether Em could reliably inform ART decisions in advance, even when semen parameters are within normal range. Importantly, by demonstrating high intra-individual variability even in normozoospermic samples, our findings establish a foundational limitation for the predictive use of Em as a standalone marker. This variability would likely be compounded in suboptimal samples, making its predictive value even more complex. We agree that extending this research to oligo-, astheno-, or teratozoospermic samples represents a valuable and necessary next step for a more comprehensive understanding of Em dynamics across the spectrum of male fertility. We have now explicitly acknowledged this in the Discussion (lines 274-279).
Reviewer 2 Report
Comments and Suggestions for Authors
This manuscript presents an interesting study investigated the temporal variability of human sperm membrane potential (Em) as a predictor for assisted reproductive technology (ART) outcomes. This study addressed a clinically relevant gap in fertility diagnostics by evaluating whether Em measurements, previously linked to IVF success, can reliably guide treatment decisions when assessed prior to ART procedures. These findings are important and significant.
Major issues:
1.​​​​ The study uses healthy donors, not infertility patients, please explain the reason.
​​2. In MATERIALS AND METHODS section, please provide details information of sperm capacitation. From figure 1 “membrane potential of sperm cells after 5 h of capacitation”, why detection Em value after 5 hrs, not 30mins, 1 hr, 2 hrs…? after 5 hrs capacitation, most of sperm experienced acrosome reaction and hyperactivities maybe decreased.
​​3. While variability is established, the manuscript minimally explores why Em fluctuates (e.g., hormonal cycles, oxidative stress…). Please expand a brief discussion of potential mechanisms.
4. please provide more details of donors’ sperm quality parameters.
Author Response
Thank you for the opportunity to revise our manuscript and for the thorough and constructive feedback provided. We deeply appreciate the time and effort dedicated to evaluating our work.We have carefully considered each comment and suggestion, and we believe that addressing them has significantly strengthened the manuscript. Below, you will find a point-by-point response to all comments raised, detailing the revisions made to the text, figures, and supplementary materials. All changes in the manuscript are tracked in the re-submitted files for ease of review.
- The study uses healthy donors, not infertility patients, please explain the reason.
Response: We appreciate the reviewer’s observation. In this study, we chose to use sperm samples from healthy donors in order to establish inter-individual variability in Em under controlled capacitation conditions. This approach allows us to characterize the temporal stability of Em in the absence of underlying pathology. While it is true that infertility patients might exhibit non normospermic samples, these are usually selected for ICSI. Samples that could be used for conventional IVF usually are characterized as normospermic samples. In addition, by defining normative parameters, we aim to provide a reference framework against which Em values from infertility patients can be interpreted in future studies. This has been added to the Discussion section (lines 274-279).
- In MATERIALS AND METHODS section, please provide details information of sperm capacitation. From figure 1 “membrane potential of sperm cells after 5 h of capacitation”, why detection Em value after 5 hrs, not 30mins, 1 hr, 2 hrs…? after 5 hrs capacitation, most of sperm experienced acrosome reaction and hyperactivities maybe decreased.
Previous studies have shown that in vitro capacitation of human sperm typically occurs progressively over several hours, with many of the hallmark changes, including membrane potential (Em) hyperpolarization, becoming evident between 3 and 6 hours of incubation in capacitating media (Baro Graf et al., 2019 and 2020, references 25 and 26 in the manuscript). We selected a 5-hour time point as it provides a robust and reproducible window in which capacitation-associated changes are fully established. The text has been appropriately adapted to reflect this (lines 151-153).
- While variability is established, the manuscript minimally explores why Em fluctuates (e.g., hormonal cycles, oxidative stress…). Please expand a brief discussion of potential mechanisms.
Response: We thank the reviewer for this thoughtful suggestion. We fully agree that a more in-depth exploration of the physiological mechanisms underlying the observed sperm Em variability would enhance the interpretation and impact of our findings. While our current data do not allow us to determine causality, we have expanded the Discussion to consider possible contributors such as hormonal fluctuations, oxidative stress, subclinical inflammation, and environmental or lifestyle factors that may impact sperm function even in normozoospermic individuals. We have expanded on this in the Discussion section (lines 261-273).
- please provide more details of donors’ sperm quality parameters.
Response: We thank the reviewer for highlighting the importance of sperm quality. As stated in our Methods section and consistent with our response to Reviewer 1, all donors were screened and selected based on normozoospermic criteria according to the WHO 6th Edition guidelines, including assessments of semen volume, sperm concentration, motility, morphology, and absence of agglutination. These evaluations were performed at each sample collection to confirm eligibility.
However, we acknowledge the reviewer's request for more detailed, individualized longitudinal data for all conventional semen parameters across the three time points for each donor. While inclusion criteria were strictly enforced at every collection, the study's primary focus was on the temporal variability of Em, and thus, the detailed quantitative values for all individual semen parameters were not systematically recorded or analyzed as primary outcomes across the entire longitudinal dataset.
Reviewer 3 Report
Comments and Suggestions for Authors
-The study cohort consists solely of healthy donors with normospermic semen parameters. However, the primary target for ART is infertile patients, whose sperm may exhibit underlying functional impairments or greater inherent variability. This limitation must be explicitly emphasized in the Discussion, and future studies incorporating infertile cohorts (particularly those with idiopathic infertility) should be recommended to validate the generalizability of the conclusions.
-Was the -48.6 mV threshold used derived from studies involving healthy donors or infertile patients? While this study observed fluctuations around this threshold in healthy donors, it did not explore potential differences in baseline Em or variability between healthy individuals and infertile patients. It is recommended to discuss this point to clarify the significance of this threshold across different populations.
-The use of a 30% coefficient of variation (CV) as the threshold for "high variability" lacks sufficient citation or biological/statistical justification. The rationale for selecting this specific threshold (e.g., reference to similar studies, clinically acceptable range?) should be provided. Alternatively, this threshold should be explicitly acknowledged as an outcome of exploratory analysis and interpreted with caution.
-Figure 1 exclusively presents the Em values following capacitation (CAP 5h). The temporal trend of Em in the non-capacitated state (NC), even if exhibiting minor variability, holds value for understanding the stability of the basal sperm state. It is recommended that this data be presented as a supplemental figure.
-The results of post-hoc comparisons for Day 0 vs. 14 and Day 14 vs. 28 (even if non-significant) should also be reported.
-References 29 and 30 cited in the Discussion are missing from the reference list.
Author Response
Thank you for the opportunity to revise our manuscript and for the thorough and constructive feedback provided. We deeply appreciate the time and effort dedicated to evaluating our work.We have carefully considered each comment and suggestion, and we believe that addressing them has significantly strengthened the manuscript. Below, you will find a point-by-point response to all comments raised, detailing the revisions made to the text, figures, and supplementary materials. All changes in the manuscript are tracked in the re-submitted files for ease of review.
1. The study cohort consists solely of healthy donors with normospermic semen parameters. However, the primary target for ART is infertile patients, whose sperm may exhibit underlying functional impairments or greater inherent variability. This limitation must be explicitly emphasized in the Discussion, and future studies incorporating infertile cohorts (particularly those with idiopathic infertility) should be recommended to validate the generalizability of the conclusions.
Response: We thank the reviewer for this crucial comment. We fully agree that the primary target population for ART is indeed infertile couples. Among these, sperm samples might exhibit unique challenges, including underlying functional impairments or potentially greater inherent variability compared to healthy donors. However, many of these sperm samples are normal according to WHO guidelines, and these samples are the ones that are usually chosen for conventional IVF. Our study was specifically designed to evaluate the temporal dynamics of sperm Em within normozoospermic samples, as used for conventional IVF procedures. The rationale for this initial approach was to establish the inherent physiological variability of Em under optimal baseline conditions, where conventional IVF is most often indicated and where Em-based testing has been previously proposed as a decision-support tool. Our objective was to assess whether Em could reliably inform ART decisions in advance, even when conventional semen parameters are within the normal range. We acknowledge that a key limitation of our current study is its focus solely on healthy, normozoospermic individuals. This important point has now been explicitly emphasized and discussed in the Discussion (lines 274-279).
2. Was the -48.6 mV threshold used derived from studies involving healthy donors or infertile patients? While this study observed fluctuations around this threshold in healthy donors, it did not explore potential differences in baseline Em or variability between healthy individuals and infertile patients. It is recommended to discuss this point to clarify the significance of this threshold across different populations.
Response: We thank the reviewer for this comment. The membrane potential (Em) thresholds discussed in our manuscript, such as the −48.6 mV threshold proposed by Baro Graf et al. [25] and the −46 mV threshold from Puga-Molina et al. [24], were indeed derived from studies evaluating in vitro fertilization (IVF) success rates. Specifically, Baro Graf et al. [25] established their threshold based on data from sperm samples obtained from clinic patients undergoing ART procedures. Similarly, Puga-Molina et al. [24] linked their threshold to IVF outcomes. This indicates that these proposed functional cut-offs are directly relevant to clinical populations facing infertility. However, it should be noticed that, as discussed above, these samples undergoing IVF procedures were normal according to WHO guidelines. Our current study, specifically focused on normospermic healthy donors to investigate the inherent temporal variability of Em under optimal, standardized conditions. While we observed significant fluctuations in Em within these healthy individuals, with changes large enough to cross these previously proposed clinical thresholds, our study design did not aim to assess potential differences in baseline Em values or the magnitude of Em variability between fertile and infertile populations. We acknowledge that a key limitation of our current study is its focus solely on healthy individuals. Therefore, we fully agree that future research incorporating infertile cohorts, particularly those with idiopathic infertility where conventional semen parameters may appear normal despite underlying functional impairments, is essential. This important point has now been expanded on in the Introduction section (lines 85-93).
3. The use of a 30% coefficient of variation (CV) as the threshold for "high variability" lacks sufficient citation or biological/statistical justification. The rationale for selecting this specific threshold (e.g., reference to similar studies, clinically acceptable range?) should be provided. Alternatively, this threshold should be explicitly acknowledged as an outcome of exploratory analysis and interpreted with caution.
Response: We thank the reviewer for raising this important point. Our selection of a 30% CV as a threshold for "high variability" in Em aiming to identify individuals exhibiting substantial intra-individual fluctuation that we hypothesize could impact functional classification. Other conventional semen parameters, such as sperm concentration, total and progressive motility, and normal morphology, are well-known to exhibit considerable biological (intra-individual) variability over time even in healthy individuals. Published data consistently show that CVs within-subjects for these parameters often exceed 30%, with reports of progressive motility around 50% and normal morphology around 58%. In this context, a 30% CV for a sensitive functional parameter like Em represents a notable and biologically relevant level of fluctuation, suggesting a significant shift in sperm's physiological state rather than minor background noise. It is important to distinguish between analytical (assay) variability and biological (within-subject) variability. For quantitative assays, acceptable analytical CVs are typically much lower (e.g., <10%) to ensure precision. The significant 30% CV we observed in Em is therefore interpreted as primarily reflecting true biological within-individual fluctuation, given that our experimental methodology was rigorously controlled to minimize technical variation. This has now been added to the Results section (lines 189-193).
4. Figure 1 exclusively presents the Em values following capacitation (CAP 5h). The temporal trend of Em in the non-capacitated state (NC), even if exhibiting minor variability, holds value for understanding the stability of the basal sperm state. It is recommended that this data be presented as a supplemental figure.
Response: We thank the reviewer for this suggestion. We agree that presenting the temporal trend of Em in the NC state provides valuable context for assessing the stability of the basal sperm membrane potential, even if exhibiting minor variability compared to the capacitated state. We have now generated a supplemental figure (Figure S1) depicting the Em values in the NC state across the three time points.
5. The results of post-hoc comparisons for Day 0 vs. 14 and Day 14 vs. 28 (even if non-significant) should also be reported.
We agree that reporting all post-hoc comparison results provides a more complete understanding of the temporal dynamics, irrespective of statistical significance. These specific values can be found in the Results section (lines 182-183).
6. References 29 and 30 cited in the Discussion are missing from the reference list.
Response: We thank the reviewer for catching this oversight. References [29] and [30] were indeed missing from the reference list due to a technical error. This has now been corrected, and both references are properly included in the updated reference list.
Round 2
Reviewer 1 Report
Comments and Suggestions for Authors
The MS has been revised and some of my comments were taken into consideration. However, the Authors did not add a table with basal semen parameters. In my opinion this table could be important to give an idea of variability of semen parameters in comparison with variability of NCEm and CEm.
Also, the Authors did not show in the figure for reviewer the association with the two most important semen parameters, i.e. motility and morphology, which also vary with time. In my opinion even if no relationships are found, these data should be added to the MS and their addition will improve it.
Overall the discussion is still, and even more, speculative regarding the possibile reasons of variability.
The Authors took into consideration my comment regarding IVF vs ICSI. I agree that IVF should be used for normozoospermic men, but reports say that ICSI is used in more than 60% of cases.
Author Response
The MS has been revised and some of my comments were taken into consideration. However, the Authors did not add a table with basal semen parameters. In my opinion this table could be important to give an idea of variability of semen parameters in comparison with variability of NCEm and CEm.
Response: We appreciate the reviewer's suggestion to include a dedicated table for basal semen parameters. We have now included Supplementary Figure S2 and Table S1, which provide the raw data for key basal semen parameters measured at each time point, specifically semen volume, sperm concentration, and total sperm number for all ejaculates included in the study, and their correlation analyses. These parameters allowed us to confirm the “normospermic” classification for all donors according to WHO criteria. Due to our laboratory's established “go/no go” sample processing workflow, which focuses on these primary parameters for inclusion, more detailed assessments such as specific motility sub-parameters or morphology were not recorded for this specific experimental setup. We acknowledge that a broader panel of semen parameters could offer additional insights, which would be valuable for future studies. This has been added to Methods, line 124.
Also, the Authors did not show in the figure for reviewer the association with the two most important semen parameters, i.e. motility and morphology, which also vary with time. In my opinion even if no relationships are found, these data should be added to the MS and their addition will improve it.
Response: We concur with the reviewer regarding the critical importance of motility and morphology as semen parameters. As noted in our response to the previous point and stemming from our standardized “go/no go” assessment for normospermic samples, detailed motility kinetics or sperm morphology were not systematically recorded for this study's specific experimental design. Therefore, we were unable to perform the suggested correlation analyses between Em and these parameters. We fully agree that investigating these associations would provide valuable insights into the multifactorial nature of sperm function and its temporal variability. This undoubtedly represents a crucial direction for future research emanating from the present findings, as these analyses fall beyond the scope and available data of this current communication. See lines 210–215.
Overall, the discussion is still, and even more, speculative regarding the possible reasons of variability.
Response: We appreciate this renewed comment on the speculative nature of the discussion regarding the reasons for Em variability. As requested in the previous revision, we have aimed to provide plausible physiological and environmental factors that could contribute to our observations. In doing so, we have carefully reviewed and adjusted the language in the discussion to frame these factors as testable hypotheses, explicitly avoiding definitive causal statements. Our primary objective with this section is to acknowledge potential influences and highlight the complex biological landscape that underlies Em regulation. We strongly believe that a key contribution of this work is precisely the identification and quantification of this significant inter-individual temporal variability in Em, which inherently opens crucial new avenues for subsequent mechanistic studies to definitively elucidate these underlying reasons. To expand on this, we have modified the Discussion, lines 272–274 and 286–290.
The Authors took into consideration my comment regarding IVF vs ICSI. I agree that IVF should be used for normozoospermic men, but reports say that ICSI is used in more than 60% of cases.
Response: We are pleased that our revision regarding the clinical relevance of Em measurements was well-received. We fully agree with the reviewer's observation that ICSI is indeed very widely employed, often even in normospermic cases. This clinical reality often stems from a desire to achieve higher success rates or to mitigate the risk of conventional IVF failure, which, as the reviewer correctly points out, does occur even with normozoospermic samples. Our study's aim is to contribute to a more nuanced decision-making process: by providing a predictive functional biomarker like Em, we aim to better inform clinicians when a normospermic sample is truly likely to succeed with conventional IVF, thereby potentially guiding more appropriate and cost-effective treatment strategies and reducing the reliance on unnecessary ICSI procedures. This has been expanded in Discussion, lines 253–259.
Reviewer 2 Report
Comments and Suggestions for Authors
The authors responded to my comments carefully and I have no further comments.
Author Response
Thank you for reviewing our revised manuscript and for your positive feedback. We are pleased to hear that our responses addressed your comments satisfactorily.
Sincerely,
Dr. Dario Krapf